# Mutational Asymmetries in the SARS-CoV-2 Genome May Lead to Increased Hydrophobicity of Virus Proteins

**DOI:** 10.3390/genes12060826

**Published:** 2021-05-27

**Authors:** Roman Matyášek, Kateřina Řehůřková, Kristýna Berta Marošiová, Aleš Kovařík

**Affiliations:** Laboratory of Molecular Epigenetics, Institute of Biophysics, Academy of Sciences of the Czech Republic, Královopolská 135, 61265 Brno, Czech Republic; matyasek@ibp.cz (R.M.); rehurkova@ibp.cz (K.Ř.); kristyna.marosiova@gmail.com (K.B.M.)

**Keywords:** SARS-CoV-2, coronavirus, mutability, evolution, genetic variation, apolipoprotein B mRNA editing enzyme (APOBEC), amino acid hydrophobicity

## Abstract

The genomic diversity of SARS-CoV-2 has been a focus during the ongoing COVID-19 pandemic. Here, we analyzed the distribution and character of emerging mutations in a data set comprising more than 95,000 virus genomes covering eight major SARS-CoV-2 lineages in the GISAID database, including genotypes arising during COVID-19 therapy. Globally, the C>U transitions and G>U transversions were the most represented mutations, accounting for the majority of single-nucleotide variations. Mutational spectra were not influenced by the time the virus had been circulating in its host or medical treatment. At the amino acid level, we observed about a 2-fold excess of substitutions in favor of hydrophobic amino acids over the reverse. However, most mutations constituting variants of interests of the S-protein (spike) lead to hydrophilic amino acids, counteracting the global trend. The C>U and G>U substitutions altered codons towards increased amino acid hydrophobicity values in more than 80% of cases. The bias is explained by the existing differences in the codon composition for amino acids bearing contrasting biochemical properties. Mutation asymmetries apparently influence the biochemical features of SARS CoV-2 proteins, which may impact protein–protein interactions, fusion of viral and cellular membranes, and virion assembly.

## 1. Introduction

The high plasticity of coronavirus (CoV) genomes allows them to adapt to new hosts and ecological niches rapidly and gives them potential as candidates for causing pandemics [1,2]. Indeed, there have been three outbreaks of coronaviruses in human populations in this century alone, while only SARS-CoV-2 has caused a pandemic (declared by the WHO in March 2020). The outbreak of COVID-19 caused by the coronavirus SARS-CoV-2 has caused more than 100 million infections and almost 2.5 million deaths worldwide (WHO report, updated 16 February 2021). Researchers around the world are continuously monitoring the genomic diversity of SARS-CoV-2 with a focus on the distribution and characterization of emerging mutations. As a result of these efforts, genome sequencing of the virus has generated a huge amount of data. Currently (16 February 2021), there are almost 500,000 full-length genomes in the GISAID database and 100,000 in the GenBank database. As the virus circulates in its host, multiple mutations appear, and some are eventually fixed. Indeed, there is already low-level genetic variation among circulating SARS-CoV-2 strains, which have arisen just one year since the virus outbreak [3,4,5,6,7,8]. There are also conflicting reports about the time of zoonotic transfer and the origin of SARS-CoV-2. Sequence data indicate that the bat RaTG13 coronavirus with 96% identity [9,10] remains the closest relative of SARS-CoV-2 and that a lineage leading to SARS-CoV-2 has been circulating in bats for decades [11]. Moreover, recombination between bat and pangolin CoVs has been proposed to explain some genomic features of the SARS-CoV-2 [12]. An unknown origin, together with high dynamics during the pandemic, seems to support a circulation model of the SARS-CoV-2 virus’ evolution [13].

Mutations in RNA viruses arrive due to three processes, and most observed mutations are neutral, although some may be advantageous or deleterious. They can arise intrinsically by copying errors during viral replication by recombination between two viral lineages, or by host RNA-editing systems, as part of host immunity [2]. As with other RNA genomes, coronavirus mutability is relatively high (1.4 × 10^−4^–10 × 10^−4^ per site per year) [14,15], making it about two fixed mutations a month. Most candidate mutations under natural selection are thought to have emerged repeatedly and independently in separate viral lineages (homoplasies) [16,17]. Mutations may alter virus structural and non-structural proteins and influence the transmission, allowing it to spread more easily, or severity, allowing it to cause more severe disease. To date, there is no or little epidemiologic evidence that any of the emerging SARS-CoV-2 mutations have caused a dramatic change in virus transmission and virulence [18,19].

Nevertheless, the D614G mutation in the S-glycoprotein has been proposed to accelerate replication and increase virion production [7]. Similarly, mutations in the viral RdR polymerase might alter its processivity and contribute to virus fitness [20]. Studies of virus mutability are important to assess the longevity of developed vaccines, especially when a large proportion of the population is expected to enter vaccination programs. Emerging mutations may also negatively influence PCR-based detection of viruses in clinical screens [21].

Data from the first year of the pandemic revealed that emerging mutations are non-random and highly skewed to C>U substitutions [3,16,22,23,24,25], which also account for the great majority of differences between bat CoV-2 RaTG13 and human SARS-CoV-2 [22,23]. These mutation asymmetries have frequently been attributed to the cytidine deaminase activity of apolipoprotein B family enzymes (APOBEC) [2,26]. Although most RNA editing mutations are detrimental or neutral, some may be beneficial and contribute to the adaptation of the virus in new hosts [27]. It is suspected that some mutations caused by RNA editing could have assisted in shaping a receptor-binding domain within the S-protein critical for SARS-CoV-2 virus host’s range and the infectivity [22]. Mutation asymmetries may also potentially account for the altered codon preferences seen in some coronaviruses, including SARS-CoV-2 [28,29]. 

The wealth of sequence data generated during the COVID-19 pandemic offers valuable material for evolutionary studies. In this report, we were asked the following questions: (i) what are the mutation profiles and frequencies of substitutions in various SARS-CoV-2 lineages differing in abundance and origin? (ii) How do emerging mutations alter the biochemical properties of residing amino acids? (iii) What are the antigenic consequences of the most common mutations in the spike (S) protein. To address these questions, we analyzed 95,000 genomes covering all major SARS-CoV-2 lineages by various bioinformatics tools. We obtained evidence that mutation spectra are stable over time, leading to an enrichment of virus proteins with hydrophobic amino acids. The consequences of these evolutionary trends are also discussed.

## 2. Materials and Methods

### 2.1. Source of Sequences

Sequences were retrieved from the Global Initiative on Sharing All Influenza Data (GISAD) website (https://www.gisaid.org/, accessed on 1 February 2021) [30] using the following filters: clade, only complete genome, low coverage excluded. Accessions labeled as “high variation” or many “Ns” were excluded. Genomes were further selected from different periods in order to cover the collection year evenly. The second filter used in the CLC genomics workbench (CLC) (Qiagen, Hilden, Germany) was set as follows: trimming 200 nt from the 3’ end to exclude variation due to different lengths of polyA tract and sequencing artifacts. Sequences with unrealistically large (>30) numbers of single-nucleotide variations (SNVs) were removed from the datasets. Only complete sequences with no or few unspecified nucleotides (Ns) were considered for the downstream analysis of variants. 

### 2.2. Analysis of Variants

In a population-level study of genetic variation, trimmed genomic sequences (Table 1) were mapped to the SARS-CoV-2 reference Wuhan-Hu-1 genome (MN908947) in CLC using the command *Resequencing analysis/Map reads to the reference*. Mapping parameters were as follows: Match score_1, mismatch cost_2, linear gap cost. The length fraction of alignment was 0.8; the required similarity threshold 0.8. SNVs were called using the Basic variant detection tool estimating no error model calling any variant satisfying the parameters. These were set as follows: genome coverage_1000; counts_50; frequency_5 or 1. The output file contained information about the position of SNV in the reference genome, nucleotide type in the reference and allele, and the frequency, count, and coverage. The file was converted to a csv format by the command *Export* in the main menu. The character of the amino acid change underlying nucleotide substitutions was determined as follows: The triplets were identified based on the annotated reading frames of the reference genome in CLC (GFF-encoded information is retained after the conversion to the clc format). In order to double check the identified amino acid changes, we introduced allelic nucleotides into the reference sequence and converted coding regions (in frame) to protein sequence using a command *Translate to protein.* The mutant proteins were aligned (pairwise) with the “wild type” reference protein sequences in the GenBank (Appendix A) and substitutions were visually checked. We also analyzed a data set of a recently published study [31] containing variants identified in a population of viruses from a single COVID-19 patient undergoing antibody therapy. The infecting strain was assigned to GR GISAID lineage (20B according to the Nexstrain nomenclature) bearing the D614G Spike variant.

Sequence contexts of mutations were identified in sequences with annotated SNVs. Proximal 5’ and 3’ bases were counted and frequencies normalized to the genome representation of each nucleotide in the SARS-CoV-2 genome according to the formula: Normalized count=Ci/(f(Ni)*4); Ci: observed count of i-nucleotide; f(Ni): frequency of i-nucleotide in the SARS-CoV-2 reference genome (A = 0.299, C = 0.184, G = 0.196, and U = 0.321).

### 2.3. Phylogenetic Analysis

Consensus sequences were obtained from mapping files using the command “extract consensus sequence” in CLC with a vote when the base was present in a majority of sequences. Consensus sequences derived from each clade and a reference SARS-CoV-2 genome (MN908947) were aligned using a Progressive alignment tool (CLC) based on Clustal W. Alignment parameters were set as follows: insertion opening cost_1, gap insertion extension cost_3 (this value was chosen to minimalize short gaps in the alignments), and deletion cost_1. Indels were not considered. A phylogeny neighbor joining tree was constructed using the Juke Kantor method implemented in CLC. Nucleotide composition was determined in virus consensus sequences from individual clades.

### 2.4. Protein Analysis

Amino acids’ hydrophobicity values were taken from the https://www.cgl.ucsf.edu/chimera/docs/ server ([32], accessed on 1 February 2021), corresponding to those experimentally determined by [33]. In addition, an alternative hydrophobicity scale was used to validate the results [34]. The hydrophobicity shifts at mutated sites were calculated as the hydrophobicity value of an amino acid in the allele minus the reference. Antigenicity plots were generated using a prediction tool implemented in the CLC program (protein analysis/antigenicity prediction) based on the algorithm of Welling et al. [35]. The method is based on calculation of the percentage of each amino acid present in known antigenic determinants compared to the percentage of the amino acids in the average composition of a protein. The index of antigenicity was evaluated over the whole S-protein using a window size of 11 amino acids.

### 2.5. Data Processing

The data files in csv format were exported to Microsoft Excel and further processed using program functions (e.g., “countif”, “sum”, and “count2”). Statistical tests were carried out using Microsoft Excel, R-studio packages [36], and web tools (Mann–Whitney U tests) [37].

## 3. Results

### 3.1. Mutation Spectra in the SARS-CoV-2 Clades

GISAID used genetic markers based on single nucleotide variation (SNV) to define eight distinct clades of SARS-CoV-2 variants, including S, L, O, V, G, GH, GR, and GV (Figure 1A). Phylogeny studies showed that the “early diverging” L, S, O, and V clades are well-separated from the “late diverging” G, GH, GR, and GV clades ([38], and Figure 1B). We analyzed genetic variation in these lineages of SARS-CoV-2 genomes collected from November 2019 until 31 January 2021. The number of analyzed genomes approximately corresponds to the clade representation in the GISAID database (Table 1 and Figure 1A). 

In each clade, variants were called at the 5% level, which means they occurred in a minimum of 10–100 genomes depending on the clade abundance (Table 1). Variant calls resulted in 135 SNVs (Appendix A), which occurred at variable frequencies (Figure 1C and Appendix A). In total, 78 substitutions were classified as non-synonymous sites while 50 were synonymous sites. Two SNVs at positions 27,972 (ORF8, Gln27) and 29,645 (ORF10, Val30) were non-sense mutations inducing stop codons. Five SNVs occurred in the untranslated regions (UTR). The most common C>U transitions were followed by G>U transversions accounting for almost 60% of all substitutions (Figure 2). Out of 78 nonsynonymous substitutions, there were 45 (58%) towards U (i.e., C>U, A>U, and G>U), 7 (9%) from U to other nucleotides, and 26 (33%) substitutions did not involve Us.

The C>U transitions accounted for most emerging substitutions ranging 33–64% between the clades. The other abundant mutations were G>U transversions (0–33%) and A>G (0–22%) and G>A (0–40%) transitions, while the remaining eight substitution types negligibly contributed to variation. There were far more C>U substitutions than the reverse U>C substitution (5:1 ratio). Similarly, the G>U substitutions dominated over the U>G substitutions (19:1), and the C>A transversions dominated over those of A>C (5:0). 

Globally, marked asymmetries exist in mutation types between the SARS-CoV-2 genomes, particularly in substitutions involving uracil (U).

Mutation spectra between early (L, O, S, and V) and late (G, GH, GR, and GV) clades were similar (Spearman’s test, Rs = 0.7343, P2 = 0.05, Appendix A). Variants called at 1% and 5% frequency levels were strongly correlated (Rs = 0.8934, P2 = 0.001). We also analyzed recently published data from a single patient derived from 23 sequential respiratory samples collected over 101 days of immunotherapy [31]. Among 100 substitutions, both the C>U and G>U type predominated, accounting for 33% and 17% of all variants, respectively (Figure 2 and Appendix A). Frequencies of individual substitution types in the global and single individual data sets were positively correlated (Figure 2B,C).

### 3.2. Biochemical Properties of Amino Acids Changed by Nucleotide Substitutions

Previous mutation analysis showed a marked (almost 6-fold) excess of non-synonymous substitutions towards U (i.e., C>U, A>U, and G>U) over those from U to other nucleotides. We addressed the question of how these mutational asymmetries influence codon amino acid hydrophobicity (Appendix A). Figure 3B depicts shifts in differential hydrophobicity values at non-synonymous sites along the SARS-CoV-2 genome. 

Out of 76 amino acid substitutions, 44 (58%) changes occurred, making amino acids more hydrophobic. A considerably lower number of substitutions, 29 (38%), resulted in opposite shifts, i.e., to less hydrophobic amino acids. Three (4%) amino acid substitutions did not influence the hydrophobicity of the site. Some changes were quite dramatic. For example, 14 substitutions increased the amino acid hydrophobicity by a factor of five (based on the Kyte and Doolittle’s scale [33]). By contrast, only three substitutions decreased the hydrophobicity by a similar magnitude. In order to validate the results, we also calculated shifts using the hydrophobicity scale of Palecz [34], which is based on averaged values obtained by different biochemical approaches (Appendix A). It provided similar results to the scale of Kyte and Doolittle [33]. Late clades showed slightly increased cumulative hydrophobicity levels on average (1.45) compared to early (1.20) clades and single (0.61) individual viruses (Figure 4) while the difference was not significant (*p* > 0.05, Mann–Whitney U test, Appendix A). We also analyzed the relationship between the nucleotide substitution type and hydrophobicity resulting from the amino acid change (Appendix A). The C>U transitions and G>U transversions lead to more hydrophobic amino acids in most cases. By contrast, the C>A and U>C substitutions were associated with decreased hydrophobicity values.

### 3.3. The Impact of Emerging Mutations on Virus Protein Domains 

To address the question of which SARS-CoV-2 proteins were most affected by the asymmetrical character of nucleotide substitutions, we analyzed hydrophobicity shifts in the ORF1a, ORF1b, S, N, ORF3, and ORF8 subregions (Figure 5). 

Among the structural genes, the nucleocapsid (N) protein showed a relatively high ratio of non-synonymous and synonymous substitutions (Appendix A) consistent with previous studies of SARS-CoV-2 genome diversity [23,39]. As expected, the majority of mutations were from hydrophilic to hydrophobic amino acids (Figure 5 and [16]) while these were not distributed evenly across the protein. For example, in a linker region between the N- and C-terminus, we identified a mutation hot spot (Appendix A, boxed) containing four C>U substitutions within a stretch of 13 amino acids (all polar, mainly serines). All four substitutions shifted the character of amino acids from polar to highly hydrophobic (Ser188Leu, Ser194Leu, Ser197Leu, and Pro199Ile). 

The mutability of the surface S glycoprotein is a critical issue in vaccination effectivity and longevity. In our data sets (Appendix A), nearly all variants (8 out of 11 non-synonymous SNVs) of this protein occurred in the GR clade suspected to harbor higher infectivity strains than other clades [7]. Therefore, we analyzed mutations within or proximal to the ACE2 receptor binding domain (Table 2), which appeared relatively recently in evolution and represents epidemiologically and clinically important variants of the virus [40,41]. 

Surprisingly, only three variants (30%) showed a moderate increase in hydrophobicity, and the overall cumulative hydrophobicity shift was negative, i.e., towards hydrophilic amino acids.

We also carried out computer prediction of the antigenicity of the reference S-protein (protein ID YP_009724390.1) and a mutant protein containing several frequently occurring substitutions in the GR clade (Appendix A). Out of eight amino acid substitutions, only two had a clear effect of antigenicity. Specifically, the Pro681His substitution (shift to a more polar amino acid) increased the antigenicity of a mutated protein (red arrow). In contrast, the Thr716Leu (shift from polar to a hydrophobic amino acid) reduced the antigenicity of the subregion (green arrow).

## 4. Discussion

Over more than one year of SARS-CoV-2 virus circulation in the human population, the virus has acquired mutations and diversified into several clades. Although we did not rigorously analyze intra-clade diversity since this may appear to be an uneasy task sensitive to population size, setting of variant thresholds, homoplasy, and perhaps other factors, we did not observe marked differences in mutation spectra across the clades. Rather, clade size correlated with the number of mutations in each lineage, suggesting that transmission frequency and population size are likely major factors contributing to the SARS-CoV-2 genome diversity. The lineages that arose early in the pandemic showed a similar proportion of major substitutions to those that arose later in the pandemic. Variants that emerged during patient treatment (plasma immunotherapy [31]) also showed similar mutation profiles. These results indicate that the mutation spectra are not influenced by the time the virus circulates in its host and the medical treatment, consistent with earlier reports obtained with relatively small genomes [22,23]. Small differences between clades can be explained by sampling and population origin [47]. 

### 4.1. Possible Mechanisms Leading to Mutation Asymmetries in the SARS-CoV-2 Genomes 

The asymmetry of nucleotide substitutions (high C>U but low U>C rates), sequence motifs flanking the mutated nucleotides [23,25], and apparent single strand affinity of cytosine deaminase enzymes [48] support the APOBEC3-mediated RNA editing mechanism of C>U conversions in SARS-CoV-2 variants [2,26]. Nevertheless, some uncertainty remains. First, not all C>U substitutions were located in preferred targets for APOBEC3 enzymes. For example, the 3’ positions flanking mutated Cs were often occupied by Cs instead of the preferred A/U in our data sets (Appendix A and Appendix A). Second, sites of reverse mutations were also frequently flanked by As, particularly at the 5‘ end [23]. Third, the serial passage of human coronaviruses in APOBEC3-overexpressing cells did not result in hypermutation in progeny viruses [49]. Therefore, alternative, not mutually exclusive, explanations have been proposed, including the spontaneous deamination of Cs [50], short palindromic sequences [51], and UV light-induced mutagenesis [3].

Relatively high G>U transversions are more difficult to interpret since this type of substitution cannot be attributed to any known RNA editing systems. These substitutions were not so pronounced in early analysis of SARS-CoV-2 variation [22,23], while more recent studies performed on larger data sets revealed G>U transversion as the second or third most frequent substitution in the mutation spectra (see Figure 2 and [3,16,52,53]). The G>U transversions could be explained by a keto-enol tautomery generating wobbling of the U base [54] and pairing with Gs instead of with A. However, coronaviruses display 3’>5’ exonuclease activity [55], which likely removes most mispaired nucleotides during RNA polymerization. An alternative explanation is the occurrence of modified nucleotides in the SARS-CoV-2 genome. For example, reactive oxygen species (ROS) may oxidize Gs to 7,8-dihydro-8-oxo-guanine (8-oxoguanine) that can also base pair with adenine (apart from canonical cytosine), yielding G-to-U transversions [26,52]. Nucleotide lesions in a positive RNA strand would lead to G>U substitutions; lesions in a complementary strand to C>A substitutions. It may be significant that the SARS-CoV-2 virus primarily replicates in oxygen-rich epithelial tissues of the upper respiratory tract, which may contain high levels of reactive oxygen species. It is tempting to speculate that viruses chronically exposed to this environment may contain oxidation products of Gs, which could contribute to mutagenesis. Interestingly, a rubella virus (harboring a single-stranded RNA genome), typically replicating in lymph nodes, does not show an elevated frequency of G>U mutations, while it does show an increased frequency of C>U mutations [25]. Indeed, experimental validation of the hypothesis is needed. In any case, relatively frequent G>U transversions might explain why G is the second (after C) least represented in a nucleotide in the SARS-CoV-2 genome (Figure 6).

The question arises as to the global impacts of mutation asymmetry (skewed away from C and G) on SARS-CoV-2 evolution. They may contribute to reducing Cs and Gs in favor of Us and influencing the evolution of coronavirus genomes [23]. Of note, we observed a slight increase in U accompanied by a decrease in C contents in genomes from late diverging clades (Figure 6). Certainly, the observed U-enrichment of the SARS-CoV-2 genome caused by C>U asymmetry cannot explain relatively large differences in U- contents between coronaviruses, and other mechanisms, such as drift, should be considered. However, because of apparent species-specific differences in APOBEC3 C-deamination enzymes [2], the U enrichment of coronavirus lineages could be due to the time the virus has been circulating in its host. 

### 4.2. Emerging Nucleotide Substitutions May Increase the Hydrophobicity of Viral Proteins 

The analysis of mutation characters revealed a remarkable trend towards hydrophobic amino acids in a spectrum of SARS-CoV-2 mutations. On average, the number of substitutions leading to a more hydrophobic amino acid was almost twice as high as those of reverse. However, there were differences between individual regions. It may be somewhat surprising that the highest number of SNVs (and concomitant hydrophobicity shifts) was found in a conserved nucleocapsid phosphoprotein (N-protein) whose primary function is to package the genomic RNA. The N-protein is rich in serine residues, particularly in the linker region. More than 80% of all serine mutations were located in UCA and UCU (bearing C>U substitution) and AGU (G>U substitution) codons, while mutations in the other three other serine codons were rare. Interestingly, the sequence contexts of C>U mutations within the UCA and UCU codons fulfill the criteria of preferential targets of cellular APOBEC3 enzymes [2]. Thus, RNA editing combined with codon preferences in the SARS-CoV-2 genome [28,29] might be responsible for the high mutation rates of the N-protein. It should be mentioned that the frequency of linker region mutations was relatively low in global sets (6–12%), consistent with a previous report [53], and haplotypes with multiple mutations were almost non-existent (Appendix A), suggesting their mostly negative effect on virus fitness.

The hydrophobic character of emerging amino acid mutations was also apparent in non-structural proteins, particularly in ORF1b, which encodes several proteins essential for virus replication. The character of substitutions in this subregion (Figure 3) indicated a marked excess of hydrophobic amino acids over those of hydrophilic (about twice of the average). Of note, a highly penetrant Pro323Leu variant (induced by C14408U substitution) in NSP12 (RNA-dependent RNA polymerase (RdRp)) has been reported to associate with increased transmissibility [53] and mutability [20] of the SARS-CoV-2 virus. Computer modelling revealed a loop of RdRp that interacts with NSP8 [53], an essential auxiliary cofactor needed for replication of long RNA [56,57]. Perhaps, increased hydrophobicity of protein domains may stabilize the replication complex, allowing faster virus replication. Hydrophobic domains are also frequent targets for inhibitors of the RdRp activity of NSP12 [58]. Better knowledge of their mode of evolution might be useful for antiviral drug design.

Differences exist between individual proteins and domains. For example, the fast-evolving S-protein showed relatively few mutations towards hydrophobic amino acids. Further, major S-protein mutations constituting variants of interest (defined according to distinct epidemiological and clinical parameters [41]) showed an opposite trend, namely shifts toward hydrophilic amino acids (Table 2). These variants completely lacked the globally abundant C>U and G>U substitutions. In contrast, the substitutions towards purines (A, G) predominated (80%). Perhaps, critical S-protein regions are under the selection pressure of antibody or coalescence plasma treatment counteracting global trends. Unusually high levels of non-synonymous substitutions (high d*N*/d*S* ratios) observed in its receptor binding domain [22,23] are consistent with the hypothesis. 

### 4.3. Relationship between the Nucleotide Composition of Codons and Amino Acid Hydrophobicity

We previously proposed that the cytosine deamination events may be responsible for some amino acid biases [22] since C-to-U (T) substitutions lead to a higher frequency of hydrophobic amino acid codons [50]. Indeed, most (90%) C>U substitutions altered codons from less to more hydrophobic amino acids in the SARS-CoV-2 genomes. By contrast, three C>U substitutions (13%) had the opposite impact; however, these involved Leu>Phe changes, which affected the biochemical property of the site only a little (both amino acids possess similar hydrophobicity values, which are even flipped in some scales). Thus, it is prudent to conclude that nearly all non-synonymous C>U substitutions increase the hydrophobicity of virus proteins. The second most abundant G>U substitution may also contribute to “drive” towards more hydrophobic amino acids. The magnitude of hydrophobicity change was, in general, lower than that of C>U. So, we are left wondering why the substitutions leading to Us often increase the hydrophobicity of an amino acid residing in the mutated site. We propose that differences in the nucleotide composition of amino acid codons (Figure 7) underlie these biases and provide the most parsimonious explanation of our observations. Particularly, variation in the second codon position dramatically influences the biochemical property of encoded amino acids. Specifically, codons with U almost exclusively encode hydrophobic amino acids, while codons with A exclusively encode hydrophilic amino acids. Biases are even more pronounced when considering amino acids with contrasting biochemical properties (Appendix A).

The codons of all four most hydrophobic amino acids (Phe, Ile, Leu, and Val) contain a U at the first or second positions. Consequently, any non-synonymous substitution toward U likely generates a codon for one of these amino acids. In contrast, a substitution from U to other nucleotides increases the chances of polar amino acids since none of the most polar amino acids (Asn, Asp, Arg, Gln, Glu, and Lys) contain a U in the first or second position. The U in the third wobbling base seems to be variable in both groups. Codons with U or A at the third position were reported to be enriched in the SARS-CoV-2 genome [29].

## 5. Conclusions

Overall, the study demonstrated that the mutational asymmetry leads to asymmetry in the amino acid composition and increased hydrophobicity for SARS-CoV-2 proteins. Hydrophobic amino acids may affect the folding of proteins and their interactions in RNA polymerase complexes, perhaps stimulating virus replication. The increase in protein hydrophobicity could have many important clinical consequences, such as prolonged mucosal colonization and persistence, minor distribution in plasma, and greater stability in aqueous cytosol. Further experimental studies are needed to confirm these hypotheses.

## Figures and Tables

**Figure 1 genes-12-00826-f001:**
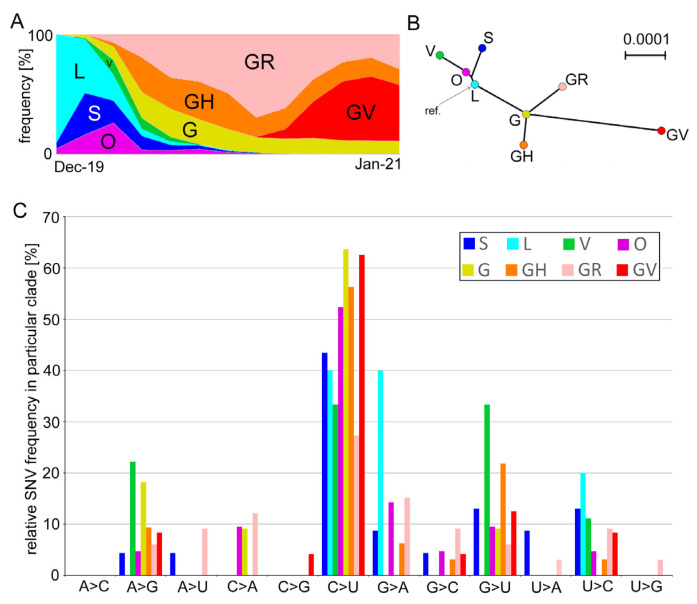
Genetic variation and character of SNVs in prominent SARS-CoV-2 phylogenetic clades. (**A**) Clade abundance evolution in the first year (taken from the GISAID web page on 1 February 2021). (**B**) Unrooted Neighbor Joining tree constructed from consensus sequences of genomes derived from individual clades. (**C**) Frequency of substitutions in individual clades (Table 1). Variants were called using a 5% frequency threshold.

**Figure 2 genes-12-00826-f002:**
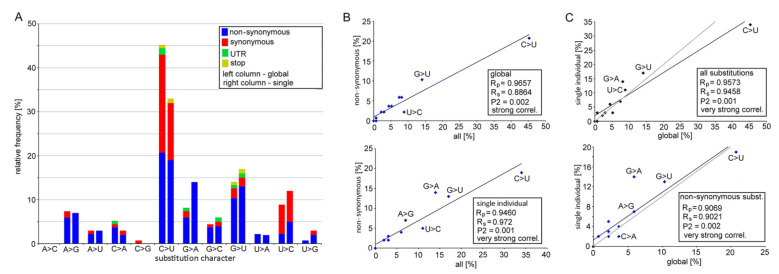
Mutation spectra of SARS-CoV-2 genomes analyzed in global data sets and virus sequences isolated from a single patient. (**A**) Relative frequency of individual types of substitutions. Left columns: the character of 135 substitutions identified in eight global lineages (Figure 1); Right columns: the character of 100 substitutions identified in virus genomes from a single individual. Data can be found in Appendix A. (**B**,**C**) The Spearman’s and Pearson’s statistics for 12 types of substitutions: (**B**) Correlation between the non-synonymous and all substitutions for global sets (upper panel) and single individual (bottom) levels. (**C**) Correlation between global sets and virus genomes from a single individual for all (upper panel) and non-synonymous (bottom) substitutions. Black lines represent linear regressions derived from measurements. Gray lines represent the equality between global data sets and a single individual (y = x).

**Figure 3 genes-12-00826-f003:**
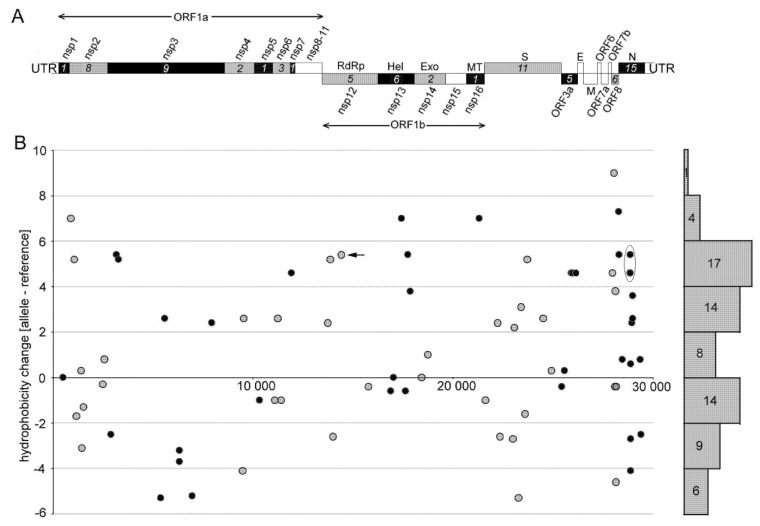
Distribution of non-synonymous substitutions along the SARS-CoV-2 genome and the character of induced amino acid changes. (**A**) A scheme of the SARS-CoV-2 genome organization. Numerals in italics indicate the number of substitutions identified within each gene. Genes with no detected substitution are in empty boxes. Nsp1-16 indicate genes for nonstructural proteins: RdRp: RNA-dependent RNA polymerase; Hel: helicase; Exo: 3′-to-5′ exonuclease; Met: 2′-O-ribose methyltransferase. Structural proteins are represented by surface glycoprotein (S), membrane glycoprotein (M), nucleocapsid phosphoprotein (N), and envelope protein (E). (**B**) Characters of amino acid hydrophobicity changes were computed using the hydrophobicity scale of Kyte and Doolittle [33] (Appendix A). For better resolution, values (circles) are shown in black or gray colors, which assign them to the same colored (black and gray) genes in panel A. Number of substitutions in individual hydrophobicity intervals is given on the right margin. Arrow—the widespread Pro323Leu substitution within the NSP12 protein. Oval—cluster of C>U substitutions underlying prominent amino acid changes in the linker region of the N-protein.

**Figure 4 genes-12-00826-f004:**
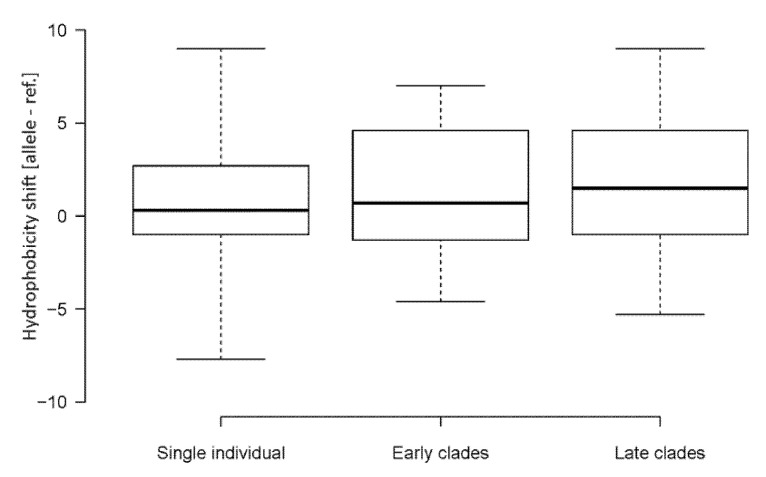
Shifts in amino acid hydrophobicity caused by SNVs in SARS-CoV-2 genomes. Box plots were constructed from data sets in Appendix A. Boxes—Q1 and Q3 quartiles. The vertical line inside the box marks the median. Whiskers extend to the minimum and maximum values. Hydrophobicity scales are according to Kyte and Doolittle [33]. Lineage groups are as in Table 1. The number of SNVs are indicated in each group. Differences between the groups were not significant *(p* > 0.05, Mann–Whitney U test).

**Figure 5 genes-12-00826-f005:**
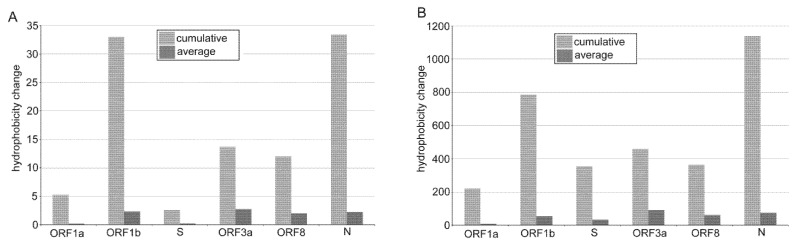
Shifts in amino acid hydrophobicity caused by SNVs in SARS-CoV-2 subregions. Cumulative and average values were computed using the amino acid hydrophobicity scales according to [33] (**A**) and [34] (**B**).

**Figure 6 genes-12-00826-f006:**
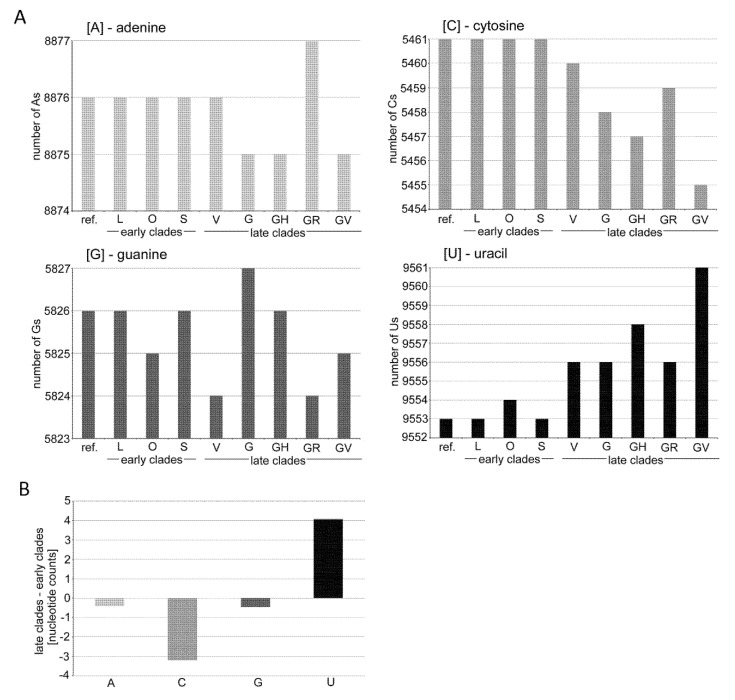
Dynamics of the nucleotide composition in the SARS-CoV-2 genomes. (**A**) Graphs rep-resent the nucleotide compositions of consensus sequences typical for individual clades. (**B**) A, C, G, and U counts. Note: the reduction of Cs in V, G, GH, and GV clades was accompanied by increases in U nucleotides. Data can be found in Appendix A.

**Figure 7 genes-12-00826-f007:**
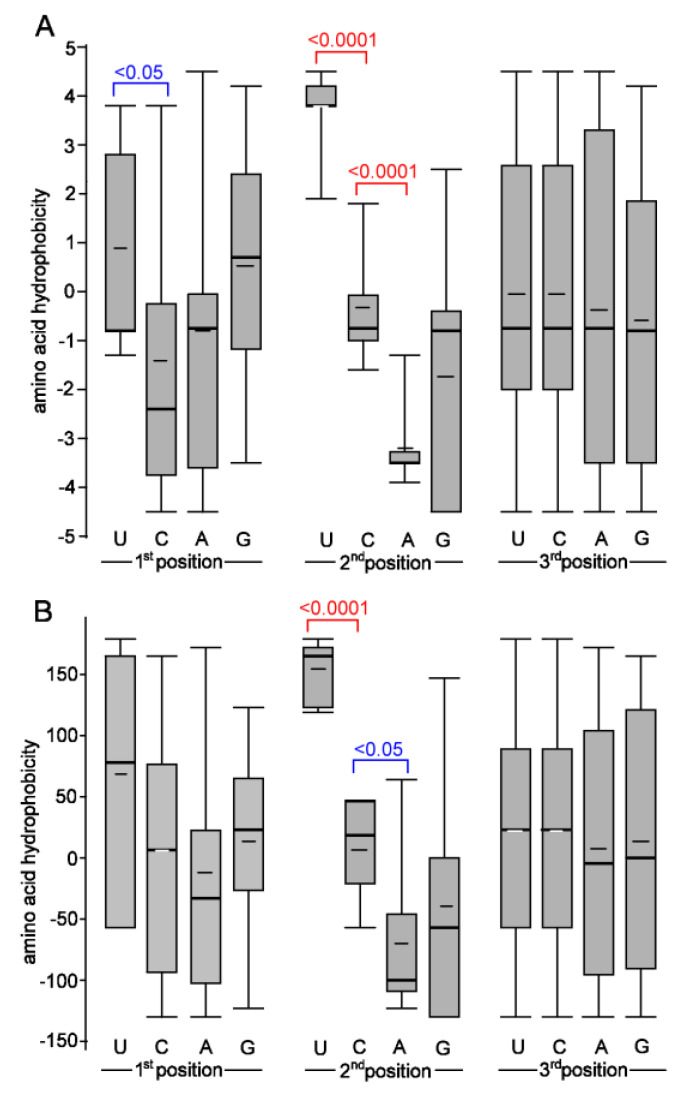
Relationship between the amino acid hydrophobicity and nucleotide composition of the codons. Hydrophobicity scales are according to [33] (**A**) and [34] (**B**). All 20 amino acids were considered. Thick and thin horizontal lines represent median and average, respectively. Differences are highlighted at the statistical level of *p* < 0.05 and *p* < 0.0001 (Mann–Whitney U test).

**Table 1 genes-12-00826-t001:** Number of genomes used in the study and their phylogenetic classification.

Group	Clade	Number of Genomes		Coverage	Characteristic Nucleotide Variations ^1^
		Total ^2^	This Study	(%)	
Early	L	4699	2222	47.3	C241, C3037, C8782, G11083, A23403, G25563, and U28144C
	O	5681	2714	47.8	G11083U, C22227U, A23403G, and G26144U
	S	7893	4532	57.4	C8782U and U28144C
	V	5320	1896	35.6	C241U, C28311U, and C23929U
Late	G	62,786	8638	13.8	C241U, C3037U, and A23403G
	GH	89,908	23,375	26	C241U, C3037U, A23403G, and G25563U
	GR	136,083	35,857	26.3	C241U, C3037U, A23403G, and A28111G
	GV	92,617	15,930	17.2	C241U, C3037U, A23403G, and C22227U

^1^ Taken from the GISAID database except for clade O, where SNVs represent mutations occurring in >30% genomes in our data set. ^2^ To 31 January 2021.

**Table 2 genes-12-00826-t002:** Biochemical characteristics of amino acids containing major S-protein variants involved virus infectivity and transmissibility.

Name ^1^	Mutation coordinate		Amino Acid Hydrophobicity ^2^			Note
	Genome Protein		Ref.	Allele	Shift	
N440K	U22882G	Asp440Lys	−3.5	−3.9	−0.4	Suspected to increase the infectivity of the virus [42]
L452R	U22917G	Leu452Arg	3.8	−4.5	−8.3	Thought to increase immune evasion and ACE2 binding [43]
S477G *	A22991G	Ser477Gly	−0.8	−0.4	0.4	Suspected to strengthen receptor interaction [44]
S477N	G22992A	Ser477Asp	−0.8	−3.5	−2.7	Strengthens receptor interaction [44]
E484K	G23012A	Glu484Lys	−3.5	−3.9	−0.4	Increased evasion from the host’s immune system [45]
E484Q	A23014C	Glu484Gln	−3.5	−3.5	0	Is suspected to increase the infectivity of the virus
N501Y	A23063U	Asn501Tyr	−3.5	−1.3	2.2	Enhances binding activity to the ACE2 receptor and is a variant of concern [46]
D614G *	A23604G	Asp614Gly	−3.5	−0.4	3.1	Dominant form in the pandemic [7]
P681H	C23604A	Pro681His	−1.6	−3.2	−1.6	Increasing prevalence worldwide [43]
P681R *	C23604G	Pro681Arg	−1.6	−4.5	−2.9	May evade the immune system [43]
Total			−18.5	−29.1	−10.6	
Average			−1.85	−2.91	−1.06	

^1^ The list is according to the ECDC variant surveillance data report [40]. Variants included in the global data set (Appendix A) are marked with asterisks (*****). ^2^ Hydrophobicity values are according to the Kyte and Doolittle scale [33].

## Data Availability

All the sequences dataset used in this study are available in the public GISAID database (https://www.gisaid.org). All data regarding results are available in the Appendix A.

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
