# Peer review of "Mutational Asymmetries in the SARS-CoV-2 Genome May Lead to Increased Hydrophobicity of Virus Proteins"

_genes, 2021, doi:10.3390/genes12060826_

Round 1
Reviewer 1 Report
This study is aimed to identify the causes and the consequences of SARS-CoV-2 genomic uracil accumulation on the line of previous publications findings. This molecular phenomenon has been associated to an increased number of triplets codifying hydrophobic residues and thus to a general increase in protein hydrophobicity.
This aspect has been investigated in previous papers (see ref 3,16,22,23,24,25) and, in order to produce an innovative work, additional analysis execution should be taken in consideration.
This study is focused on three main questions stated at the beginning of the paper (line 93-95) but actually they remain partially or poorly answered and thus, in order characterize the entity and the consequences of this genomic trend, more informative and extensive bioinformatics and statistical analysis are required.
These my major comments:
1) In order to provide a reproducible workflow it’s suggested to precisely describe followed analysis in any points.
Triplets identification and extraction methods is not distintly described, it looks like codons are extracted regardless frame-shift but by only considering mutation’s flanking nucleotides. (see rev120b). Some extra-step are been mentioned but not explained, for instance mapping step and variant calling follow FASTQ file processing for variant calling but this process is not applicable to FASTA file obtained by GISAID, thus more precise explanation is here required (see 118).
The steps leading from a GISAID alignment to a MsExcel sheet and how the authors combined mentioned commands to obtained variants calling among sequences was not mentioned; a transparent description of used bioinformatics tool is required to allow data reproducibility. A clearer description about methods behind codon annotation in table S4 is required.
2) This study lacks proper normalization and statistical tests.
Is not clear how normalization methods proposed at line 122 and normalization formula proposed in
Count=Ci/(f(Ni)*4) could adjust frequency comparison among groups, a deeper explanation is required. Given that a consistent difference between early and late clade sizes has been observed, in order to reduce eventual uneven sampling bias, a normalization or down-sampling step among groups is advised.
Results obtained by residue hydrophobic distribution analysis could be greatly supported by statistical analysis among groups (eg. hydrophilic mutations frequencies against hydrophobic ones); furthermore, additional test could be performed to evaluate the significance of U’s distribution increment over time.
3) Since this biological phenomenon has already been extensively investigated in different previous publications (eg. Rampant C→U Hypermutation in the Genomes of SARS-CoV-2 and Other Coronaviruses: Causes and Consequences for Their Short- and Long-Term Evolutionary Trajectories. P. Simmonds) it would be interesting evaluate downstream biological and clinical effects as for instance:
To assess eventual effect of hydrophobicity shifts on protein solvation, the increase of protein hydrophobicity could have many important clinical consequences such as prolonged mucosal colonization and persistence, minor distribution in plasma, greater stability in aqueous citosol etc, this information could greatly support COVID-19 characterization.
Determine domain trans-membrane regions mostly affected by these changes; it is possible to identify eventual gain or loss in trans-membrane protein stability which translate in protein functional gain or loss.
Detect eventual changes in RNA structure stability following uracil’s accumulation in genome could also reflect advantages acquired in terms of replication efficacy.
Minor points:
120a Indels were not considered.
Why have they been excluded? Indels could greatly alter nucleotides proportions and thus is an aspect worth of investigation. Furthermore those kind of genomic variations are becoming characterizing signature for emerging lineages.
120b Sequence contexts of mutations were identified in files containing the extracted triples bearing a mutated base in the middle.
Please provide a deeper explanation of codon detection process since mentioned method seems to always be considering mutated nucleotide as being the middle of the codon while it could actually be the first or third according to translation frame which can be extracted from a SARS-CoV-2 reference Genbank/GFF3 file.
122 Counts were normalized to genome representation of each nucleotide in the SARS-CoV-2 genome.
It’s suggested to better describe this normalization process goal and steps.
Normalization process is generally used to minimize artificial biases introduced by uneven clade sizes and make them comparable. Why is here used to normalize uneven frequency nucleotides counts?
135 The analysis of mutation characters revealed remarkable trend towards hydrophobic amino acids in a spectrum of SARS-CoV-2 mutations.
It could be informative to support inferred hypothesis with statistical tests taking in account the significance of observed biochemical property shift along with its increment over time (eg. non parametric test and test for trend)
143 2.5. Data processing-The data files in the csv format were exported to MsExcel and further processed using program functions (e.g., “countif”, “sum”, “count2”)
Input data structure and SNV detection method should be well described in order to make the analysis reproducible. Please provide a detailed workflow.
186 Out of seventy-six amino acid substitutions 44 (58%) changes occurred in direction from less hydrophobic to a more hydrophobic amino acid.
It would be interesting to confirm observed by applying a non-parametric U-test.
Figure 3 In Fig 3, the top bar representing SARS-CoV-2 genes is not well aligned with underlying dot chart and thus is not clear where the plotted mutations lay, moreover this bar lacks of many genes (see M,E some ORFs) that are indeed excluded from chart representation.
151 - Table 1 It’s suggested to use a standard delimiting system and replace commas with dots when delimiting thousands.
130 A phylogeny Neighbour joining tree was constructed using the Juke Kantor algorithm implemented in CLC. Nucleotide composition was determined in virus consensus sequences from individual clades. Please report mentioned tree.
117 Mapping parameters were set in order to minimize short gaps in the alignments: insertion opening cost_6, insertion extension cost_1 and deletion cost_1. Mentioned setting penalize the gap opening but not its extention; to better penalize small insertion only gap-extention score should be increased (eg 2)
Reviewer 2 Report
In this article, the authors analyzed the mutations in SARS-CoV-2 genomes. They confirmed the strong bias toward C>U mutations and more surprisingly, that if the mutations are non-synonymous, the mutated amino acids are more hydrophobic in majority. The introduction and discussion are quite interesting.
The authors have a thorough knowledge of the state of the art of genome mutations and the questions of this article were clearly asked. However, I was wondering if these questions are different enough from the other article published in Genes in 2020 where C> U transitions and hydrophobicity were also discussed: I am not sure what is the new conclusion? Is it only a confirmation of previous results on more data?
Overall, I was also missing a statistical analysis of the results, and I believe that a structural analysis of the localization of amino acid mutations could be very insightful.
Also, the quality of the English has at times hampered my comprehension of the text, even if I am not a native English speaker.
Here are my more specific comments:
Line 42 and 43: I believe that no reference to any figure should be made in the introduction. To my opinion, this information is not very necessary.
Line 40: this is the first mention of GISAID, it should ne explained.
Figure 1: For unrooted trees, I prefer a radial representation
Line 62 : “Mutations in RNA viruses arrive as a result of three processes and most are neutral, although some may be advantageous or deleterious “ We don’t study all possible mutations, only those we can observe. It is quite probable that most occurring mutations are not even observed because they are too deleterious. I would therefore add “observed mutations” after “most”.
Line 73: which parameters ?
Line 113 and after (Analysis of variants) : could you mention which algorithm was used for the pairwise comparison? I am not convinced by the listing of all these parameters without an explanation of the method.
Line 130: Could you justify your choice of the NJ method for the phylogenetic reconstruction? There exists better methods now.
Line 138: could you explain what are antigenicity plots ? and what is the prediction tool you used ?
Line 149: roughly is imprecise
Table 1: proportions would help to compare the number of genomes of this study vs. the total.
Line 155: It is surprising that 5% threshold is independent of the number of genomes; usually, the sample size is important to fix a threshold.
Line 165: A<C should be A>C, rigth ?
Figure 2: the proportion of non-synonymous does not seem to follow the overall distribution, but a statistical test is necessary to assert it. A figure with arrows of different size depending on the frequencies between A, U, G, C would help to understand the results.
Line 194: “Considerably lower” is also not precise enough. A statistical test would be welcomed.
Line 206: Again, “It is evident” is also not very convincing.
In this paragraph, a (statistical) analysis of the genetic code is necessary: the hydrophobic amino acid genetic code seems to contain more U than other codons. Maybe it is the source of the observed bias? (see joined figure)
Figure 4: I am not convinced by the relevance of the figure.
Line 214 and further: impact of emerging mutations…
The mutations >hydrophobic are quite surprising in a linker.
An analysis of the structural localization would be quite interesting here.
Line 234: this part need to be explained in more details
Line 259: It seems contradictory that “mutation spectra are not influenced by the time the virus has been circulating in its host 256 and the medical treatment” but that “There is also some evidence that certain virus genotypes may 259 be influenced by the age of infected individual and a human race”. To my opinion, the latest is more a geographical effect than a real link between race or age and the virus genotype.
Line 297: I don’t understand what are those 68 (56+12) substitutions. Are they from the 78 non-synonymous mutations ? Why 10 of them have been discarded ?
Line 345 and further: The conclusion is very short and not very convincing.

Round 2
Reviewer 1 Report
The authors have satisfactorily responded to all questions and made the necessary changes to the manuscript.